# PCGen: A Fully Parallelizable Point Cloud Generative Model

**DOI:** 10.3390/s24051414

**Published:** 2024-02-22

**Authors:** Nicolas Vercheval, Remco Royen, Adrian Munteanu, Aleksandra Pižurica

**Affiliations:** 1Research Group for Artificial Intelligence and Sparse Modelling (GAIM), Department of Telecommunications and Information Processing, Faculty of Engineering and Architecture, Ghent University, 9000 Ghent, Belgium; 2Clifford Research Group, Department of Electronics and Information Systems, Faculty of Engineering and Architecture, Ghent University, 9000 Ghent, Belgium; 3Department of Electronics and Informatics (ETRO), Faculty of Engineering, Vrije Universiteit Brussel, 1050 Brussel, Belgiumadrian.munteanu@vub.be (A.M.)

**Keywords:** point clouds, autoencoder, variational autoencoder, vector-quantized variational autoencoder, real-time computing

## Abstract

Generative models have the potential to revolutionize 3D extended reality. A primary obstacle is that augmented and virtual reality need real-time computing. Current state-of-the-art point cloud random generation methods are not fast enough for these applications. We introduce a vector-quantized variational autoencoder model (VQVAE) that can synthesize high-quality point clouds in milliseconds. Unlike previous work in VQVAEs, our model offers a compact sample representation suitable for conditional generation and data exploration with potential applications in rapid prototyping. We achieve this result by combining architectural improvements with an innovative approach for probabilistic random generation. First, we rethink current parallel point cloud autoencoder structures, and we propose several solutions to improve robustness, efficiency and reconstruction quality. Notable contributions in the decoder architecture include an innovative computation layer to process the shape semantic information, an attention mechanism that helps the model focus on different areas and a filter to cover possible sampling errors. Secondly, we introduce a parallel sampling strategy for VQVAE models consisting of a double encoding system, where a variational autoencoder learns how to generate the complex discrete distribution of the VQVAE, not only allowing quick inference but also describing the shape with a few global variables. We compare the proposed decoder and our VQVAE model with established and concurrent work, and we prove, one by one, the validity of the single contributions.

## 1. Introduction

In the wake of strikingly realistic randomly generated 2D images [1,2], there is a mounting expectation for generative models to replicate the same success when automatically synthesizing 3D objects. Demand for such models arises in various domains, ranging from rapid design and prototyping for the manufacturing sector to the entertainment industry [3,4,5,6,7]. Hence, the research focus is shifting towards 3D deep generative models, as rich and flexible 3D representations are rapidly emerging. Many of these applications benefit from or require quick generation of the desired 3D object. One example is a machine learning model that fabricates levels in a video game to grant the user a unique experience [5,6]. Real-time computation allows the random generation of items or characters during the play. Tweaking the design of a 3D-printable object [7] may also require multiple iterations of a generative model and would also benefit from quick inference. Any application in extended reality, such as generating human models from motion capture data [4], requires the model to run at the same frequency as the frames. This requirement prevents the 3D generative model from widespread use in the above applications because popular generative models, such as transformers, are not parallelizable.

Point clouds are native to 3D laser scanning (LiDAR) and, therefore, are becoming ubiquitous while their processing remains challenging. In the last decade, machine learning research made considerable progress in processing point clouds for, among other things, classification and segmentation [8,9].

The point cloud representation is suitable for generating surfaces because it requires many fewer points than voxels for a comparable level of detail and, therefore, a lower theoretical bound on complexity. It is especially promising for virtual and augmented reality applications (AR/VR), as they typically have real-time constraints.

Point cloud autoencoders compress a point cloud into a lower-dimensional encoding that we refer to as a codewordDepending on the architecture, a decoder reconstructs from the codeword the input points individually or the general shape by generating an arbitrarily dense point cloud: the latter, usually called generators, suit downstream applications requiring flexible output. Autoencoders and, in particular, variational autoencoders (VAEs) [10] were the earliest method proposed for point cloud generation [11,12]. Point cloud VAEs also found applications in shape interpolation [13], feature and transfer learning [14,15], and shape inference [16]. They offer great potential for prototyping. One reason is that the compressed codeword expresses the point cloud’s global semantic variables. This property allows for a compact data representation and facilitates data exploration and conditional generation. Another reason is that they are fully parallelizable and allow quick inference. Unfortunately, posterior collapse, a known issue for variational autoencoders, prevents them from competing with the state of the art in generation quality.

Recent works [17,18] have replicated the success of vector-quantized (VQ) variational autoencoders (VQVAEs) [19] in 3D shape generation, setting the new state of the art. VQVAE proved very effective at encoding different types of grid-like data, such as images [19], into discrete latent variables whose value corresponds to the index of a dictionary, named a codebook, of learned semantically meaningful vectors. They are not affected by the blurriness and posterior collapse issues [19] typical of VAE models. Still, they need a second autoregressive model to approximate the complex discrete latent distribution to generate new samples. Because of that, they lose some advantages of variational autoencoders, such as parallel processing of the codes and data exploration through global semantic variables. For point clouds, the generation time is in the order of seconds [17], making these models impractical for AR/VR.

We propose a novel VQVAE-based point cloud generative model with an innovative sampling strategy that allows high-quality generation in real time thanks to parallel computing. Our main contribution is threefold: first, we introduce a new architecture for a decoder that processes the encoded shape in a radically new way, yielding superior reconstructions. Secondly, we propose a filter on the decoded point cloud that reduces the gaps naturally occurring when generating the points’ coordinates in parallel. Thirdly, we introduce a VQVAE model with a new sampling technique for the discrete codes based on a second smaller VAE model that allows parallel generation, reducing computation time. The following paragraphs give more context and detail to each contribution.

Our generator translates codewords into mappings from a sampling space to the Euclidean space and uses these mappings to process the points of the reconstructed shape in parallel. Differently from previous work, which folds [20], tears [21] or patches together [22,23] 2D structures, we sample the initial position of the points from a higher dimension. When ultimately projecting them in Euclidean space, our mapping describes surfaces as thin regions of space, letting the model learn to concentrate the resulting distribution along the desired shape. This solution solves topological problems and artefacts that complex shapes may cause.

A crucial difference in the architecture of our generator compared to previous work is how the codeword and the sampling are joint, therefore, how the mapping is defined. The standard way of producing the mapping adds the codeword as semantic features to the initial spatial coordinates through concatenation [20,21,22,23].

Instead, we convert the initial sampling into a high-dimensional random generator with a learned covariance and compact support. We then formulate a tensor product between the output of the random generator and the codeword. Geometrically, the codeword defines a bounding box of the resulting distribution’s support, expanding the directions most relevant to the encoded input shape and contracting the irrelevant ones. Furthermore, we extend the model by reconstructing the point cloud as a learned interpolation of different point cloud representations with an attention mechanism. This solution trades computation time for performance while avoiding known problems with glueing different mappings [21]. We show experimentally that the proposed solutions improve CAD models’ autoencoder reconstruction on the Shapenet dataset [24].

Next, we turn our attention to mitigating the effects of the sampling inconsistency, which may cause gaps in the decoded surface, increasing the loss and degrading its gradients. We introduce a new filtering approach based on Laplacian filtering and apply it on the recovered surface to avert this occurrence. Our filtering approach significantly improves the homogeneity of the recovered surface compared to the commonly employed Laplacianfiltering [21,25]. Moreover, the proposed filter adds little overhead and is suitable for similar generation methods.

Finally, we address the main computational bottleneck of the VQVAE model. We replace the autoregressive sampling of the discrete latent space with a second variational autoencoder. By doing so, we generate discrete codes in parallel and reduce the overall inference time needed for generation. To make it even more robust, we use the vectors in the VQVAE codebooks as the latent variables’ embedding. The continuous latent space of the second encoding has a tighter bottleneck, which allows for data exploration techniques that are usually not possible in a VQVAE, such as disentanglement. Different metrics show that our random generation is up to the standards of competing models with much inferior computational cost.

In summary, this paper introduces a fast VQVAE for point clouds. The three main contributions of this work are the following:A flexible architecture for a point cloud generator that improves speed, memory and performance over comparable generators. The main novelties are how the codeword determines the reconstructed shape and the use of components as a more robust alternative to patches for describing complex shapes.A filter acting on the reconstructed surface to mitigate sampling errors that affect generators. The proposed filter further boosts the reconstruction quality and can be used with other point cloud generators.A novel double encoding system where we couple the VQVAE model with a second variational autoencoder that learns the distribution of the discrete codes. Together the two autoencoders act as just one autoencoder, effectively combining the sharpness of the VQVAE with the fast generation and interpretability of the plain VAE.

A direct comparison with alternative generative models shows that our model is competitive with the state of the art in terms of standard metrics. Furthermore, we achieve realistic, diverse generated samples at a fraction of inference time compared to similarly performing models in the literature.

The paper is organized as follows. The next section lays out the background our model builds upon and reviews recent related works and the current state of the art. Section 3 presents the proposed model, with a dedicated subsection for each of the three main contributions: the decoder architecture (Section 3.1), the filter (Section 3.2) and the double encoding (Section 3.3). The last subsection (Section 3.4) describes the encoder and the general training scheme. Section 4 has two subsections dedicated to the experiments. Section 4.1 defines the metrics to test autoencoder reconstruction and realistic generation quality. Section 4.2 compares the results of the experiments with established and concurrent work. Conclusions are in Section 5.

## 2. Related Work

Current point cloud feature extraction and segmentation strategies often alternate order-invariant methods such as dense layers acting on channels, symmetric functions such as max pooling and local operators defined on neighbourhoods calculated at runtime, effectively exploiting the flexibility of a high-dimensional channel space and the points’ local covariance [8,9]. On the other hand, strategies for point cloud generation are more diverse because they combine point cloud processing approaches with probabilistic modelling. In particular, the VAE and VQVAE probabilistic models assume a relevant role in this field thanks to their flexibility, allowing us to build on previous work with deterministic autoencoders.

### 2.1. Variational Autoencoders and Vector-Quantized Variational Autoencoders

Variational autoencoders (VAEs) [10] assume that a latent variable *z* with a known distribution p(z) governs the dataset distribution p(X)=p(X|z)p(z). A neural network functioning as a decoder approximates the conditional mean of p(X|z), learning the parameters through backpropagation. A stochastic neural network functioning as an encoder approximates the intractable p(z|X) through amortized variational inference. The core architecture is thus an autoencoder that optimizes itself through self-learning, identifying the latent variables with the codes. Minimizing the Kullback–Leibler divergence between the approximate and true latent posterior and the reconstruction error increases a lower bound on the maximum likelihood (ELBO). The prior distribution p(z) may be fixed or learned. Ref. [26] introduces a variational mixture of posterior (VAMP) prior [26]. Their encoder learns $N_{u}$ representatives ${\left(u_{r}\right)}_{r\in \{1,\ldots ,N_{u}\}}$ of the training dataset called pseudo-inputs and sets the mixture model of their posteriors $p\left(z\right)=\sum_{r=1}^{N_{u}}p\left(z|u_{r}\right)$ as prior, optimizing in parallel the prior and posterior.

VAEs retain some advantages compared to other generative models. Differently from adversarial models, they are guaranteed to cover the whole dataset distribution; in contrast to autoregressive models, they process global variables in parallel, and, unlike flow and diffusion methods, they have complete freedom in the design of the generative network. One main drawback is a tendency to experience a posterior collapse in the latent space, which happens when the regularization of the latent space necessary for realistic generation prevents it from encoding high-frequency details. As a consequence, the decoder learns to produce overly smoothened point clouds. SetVAE [27] introduced a hierarchical VAE model for point clouds, which is order-invariant with respect to the codes thanks to a suitable transformer module, and combined good reconstructions, diverse generation and efficient implementation.

Another solution to posterior collapse is adopting a discrete distribution. In particular, VQVAEs [19] proved highly successful in encoding grid-like data in codes with discrete delta distributions. The encoding is deterministic: it removes the stochastic noise that VAEs typically add to model distributions. Furthermore, it removes the $D_{KL}$ regularization term because the divergence between a posterior delta distribution and the uniform categorical prior is constant.

The posterior of a discrete code corresponds to an index from a “codebook”, a list of learned vectors called embeddings which store complex semantic information. The decoder takes in the embeddings corresponding to the predicted indices and processes them to estimate the initial input. Discrete sampling does not allow backpropagation. As an approximate solution, VQVAEs copy the loss gradients of the decoder’s input and apply them directly to the encoder output. This solution only works when the encoded features are similar to the retrieved embeddings. Therefore, a natural choice for inferencing the latent code is selecting the index corresponding to the embedding closest to the encoder. VQVAEs also add a commitment loss given by the mean square distance between the two to tie the encoder to the existing codebook. They also use the same distance as an embedding loss to learn the embeddings. Alternatively, they use a moving average update of the corresponding encoded feature.

Because the latent prior is not enforced, the aggregate latent posterior distribution is unknown. VQVAEs approximate this distribution with a second autoregressive model, allowing random generation. At the same time, the autoregressive model is not parallelizable and cannot capture different sources of variability in global variables as vanilla VAEs do. We show that a second VAE is a viable alternative to the autoregressive model that keeps all the advantages of the VAE model.

### 2.2. Generative Models for Point Clouds

With random generation, we refer to a model’s ability to approximate the dataset distribution and sample from it. In this field, standard datasets contain a specific class of model objects, including aeroplanes, chairs and cars. Representing very different objects is complicated because most semantics (for example, the ones differentiating an armchair from a seat) only make sense within a category. Therefore, most related work tests the generative method with these three classes separately by training a different model for each class.

Point cloud random generation often combines the maximum likelihood approximation of the dataset distribution with a probabilistic description of the point cloud itself, seen as a sparse distribution on $R^{3}$ where the points likely lie on the surface of the represented object [28]. The advantage over early work [11] with a deterministic description of the point cloud given by a fully connected neural network is that it allows sampling an arbitrary number of points from each point cloud and recycling the weights without having to infer the position of every point separately. Most recent models use codes to define a transformation of samples from an initial shared distribution to the target shape. This idea is particularly suitable to probabilistic frameworks such as diffusion models that first iteratively scatter the target shape to match a three-dimensional Gaussian and then learn to denoise it until recovering the target shape. Related to them, ShapeGF [28] is an energy-based model that moves samples from an arbitrary initial distribution in Euclidean space according to the gradients learned from the target shape using a denoising process. Flow models approximate a bijection transformation from a Gaussian to the desired shape. PointFlow [29] proposes a continuous normalizing flow whose theoretical and experimental estimation SoftFlow [30] improves by perturbing the target shape.

All the cited probabilistic models complement their generative part with a (variational) encoder that helps the diversity of the generated point clouds. Alternatively, SP-GAN [31] uses a generative adversarial model that learns some correspondence of realistic shapes to a 3D sphere. Autoregressive models on points [32] also exist but currently are very slow. Except for Discrete Point Flow, which takes only a few milliseconds to generate a point cloud [17,33], the generation times of all the models mentioned here are in the order of tenths of a second or more, and thus incompatible with real-time constraints [17].

### 2.3. VQVAE Models for Point Cloud Generation

Since VQVAEs do not have restrictions on the core autoencoder model, it is possible to consider various alternatives. Recent works [34] showed that learning a Signed Distance Function (SDF) leads to unparalleled point cloud reconstruction quality. High resolution requires substantial computational time [34]. AutoSDF [18] introduces a VQVAE model for a 3D-autoregressive SDF generation using Transformers. Similarly, ShapeFormer [35] trains a VQVAE model to generate a deep implicit function over the Euclidean Space. The 3D formulation of these models inevitably impacts their inference time. Closer to our work, ref. [17] proposes a VQVAE model for point clouds. First, inspired by [36], they establish a bilinear map from a shape to a sphere divided into patches. They order the patches following a spiral and recover the equivalent patches on the input. Then, they encode each input patch separately and learn the distribution of the encodings with an autoregressive model. Serializing the patches poses some limitations: nearby patches may be indexed away from each other and the bilinear mapping is only possible for shapes topologically equivalent to a sphere. Our approach is to use a global encoding instead and separate the codeword into semantic parts (codes). Instead of a slow autoregressive model, we generate the codes with a small variational autoencoder.

### 2.4. Parallelizable Decoders and Laplacian Filtering

VAEs and VQVAEs do not use the encoder during generation. Therefore, the inference time is only determined by the decoding part of the autoencoder. Fast decoders for point cloud reconstruction process all points in parallel. They typically have a geometrical approach, modelling the shape as a predominantly smooth surface. FoldingNet [20] learns to fold a square grid around the shape, often running into topological constraints. TearingNet [21] proposes a solution to allow a tearing of the grid, which allows more flexibility. AtlasNet [22] covers the target surface with multiple patches, i.e., mappings of sampled points from a plane. AtlasNetv2 [23] comes in two versions: AtlasNetv2 with patch deformations learns a high-dimensional embedding for the bi-dimensional patches; AtlasNetv2 with point translation learns patches from discrete elementary structures, learned sets of points which act like moulds and yield a fixed number of points. As in the previous version of AtlasNet, they process the patches independently and learn to estimate their relative position only indirectly from the reconstruction loss. This process may cause artefacts and misalignment along the borders of the patches [21] and is too unreliable for randomly generated point clouds.

TearingNet [21], following [25], also implements Laplacian filtering after the point cloud reconstruction because of its well-known smoothening effects, improving their model slightly. They capture the proximity of points $x_{j}$ and $x_{i}$ in an adjacency matrix with exponential weights
(1)$$\;W\left(i,j\right)=\left\{\begin{array}{cc} \exp\left(-\frac{\left|\right|x_{i}-x_{j}{\left|\right|}_{2}^{2}}{2\epsilon }\right)\; & \mathrm{if}\;||x_{i}-x_{j}{\left|\right|}_{2}\leq r\\ 0\; & \mathrm{if}\;||x_{i}-x_{j}{\left|\right|}_{2}>r, \end{array}\right.$$
thresholded on an estimated value *r* and select the unnormalized graph Laplacian
(2)$$\Delta \left(i,j\right)=\left\{\begin{array}{cc} -W(i,j)\; & \mathrm{if}\;i\neq j\\ \sum_{j}W\left(i,j\right)\; & \mathrm{if}\;i=j. \end{array}\right.$$

The filter acts on a point cloud by multiplying it by (1−λΔ) with λ=0.5, spreading the curvatures along the surface. Such a filter may help the autoencoder reconstruct smooth shapes but it is unsuitable for more complex ones. Indeed, the filter has a marginal impact on the overall metric [21]. Unlike previous work, we select a negative λ to spread the close points along the surface to cover it more uniformly.

We remark that Laplacian filtering does not remove points and has a different purpose from filtering strategies used for noise reduction in point cloud acquisition. Reducing noise during the acquisition phase generally consists of removing isolated points or groups of sparse points (noise clusters) resulting from an incorrect sensor measurement [37] or objects that are not relevant to the scan [38]. In this work, we seek to reduce noise from a source that requires a different approach.

The noise in an autoencoder’s reconstruction is caused by the degradation of the features during compression. While there could be isolated points, these, for the most part, are not mistakes but part of the risk-minimization strategy that the model uses when it is not confident that an area is empty. Attempting automatic point removal in these low-density areas is bound to the risk of removing part of the target shape. The most significant part of the noise in reconstruction results either from the inability to recover the correct shape of the surface, which is what traditional Laplacian filtering tries to mitigate by making the shape smoother, or from surface sampling discrepancies, which is what the proposed filtering approach addresses.

## 3. Methods

Our main contribution is a fully parallelizable VQVAE for real-time point cloud generation. To make parallelization possible while producing high-quality samples, we propose advancements of the VQVAE [19] design both specific to this field and applicable to other data types. We focus on three main aspects: the core point cloud autoencoder architecture, the compression of the latent space into discrete variables (quantization) and the strategy for sampling the discrete variables that allows random generation. The corresponding contributions independently build to the proposed model’s full fruition. We refer to Figure 1 for a high-level view of the model.

We present a new architecture for the generator and filtering method in Section 3.1 and Section 3.2. Section 3.3 explains our quantization procedure and introduces a variational autoencoder for discrete codewords (w-autoencoder). We propose its use as an effective sampling strategy for the complex distribution of the VQVAE latent space. Section 3.4 details our chosen training scheme and the architecture of the point cloud encoder (pc-encoder) used in the experiments.

### 3.1. Generator

Our generator models a point cloud as independent samples from a shape distribution, keeping the simplicity and the parallel processing of the geometrical approach [20,21,22,23] but using the probabilistic interpretation of other generative methods [28,29] to avoid topological and regularity issues. It takes a quantized codeword $\bar{w}$ of length $L_{w}$ in and maps an initial distribution into the codeword embedding space before projecting the distribution to the 3D space. We break it down into four blocks. In point sampling, we learn a starting distribution in a high-dimensional space. In code mixing, we propose a new layer that endows the starting distribution with the semantic meaning of the incoming codeword. We also introduce the attention and component blocks to extend the immediate projection to the Euclidean space by adding an attention mechanism. Figure 2 gives a detailed illustration of the autoencoder architecture.

Point sampling starts by sampling M points $\left\{x_{i}^{0}\right\}={\left\{x_{i}|x_{i}\in S^{N_{s}-1}\right\}}_{j\in \{1,\ldots ,M\}}$ uniformly from the hypersphere $S^{N_{s}-1}$ in $N_{s}$ dimension. While other distributions are possible, we choose one defined on the hypersphere because it has compact support, which makes tracking down the mapping of the points (see Appendix A) easier. Then, it embeds the distribution in the codeword space with a point convolution network, letting the model learn a correlation between the features and obtaining $\left\{x_{i}^{1}\right\}$.

Code mixing takes as inputs the codeword $\bar{w}$ and the previous distribution. Instead of concatenating them to the codes as commonly performed in the related literature, we multiply $\left\{x_{i}^{1}\right\}$ term by term with the codeword: we first cap the obtained points between −1 and 1 (Hardtanh); then, we use the tensor product ⊙ to generate a high-dimensional representation $\left\{x_{i}^{2}\right\}$ of the point cloud:(3)xi2=Hardtanh(xi1)⊙w¯,Hardtanh(x)=−1ifx<−1−xif−1≤x≤1,−1if1<x.

We merge the input codeword $\bar{w}$ and $\left\{x_{i}^{1}\right\}$ in a non-linear way to enable a complex interaction between the two: the codeword stretches the marginal distributions of the relevant semantic dimensions and uses the learned correlation between the $\left\{x_{i}^{1}\right\}$ points as a base for the distribution of the resulting $\left\{x_{i}^{2}\right\}$. We remark that the standard solution of concatenating the sample and the codeword determines a linear interaction in the following point convolution, which applies the non-linearity only after merging the two.

Before explaining the proposed component and attention blocks, let us consider a basic design of our architecture where we immediately obtain the output point cloud by projecting $\left\{x_{i}^{2}\right\}$ into the Euclidean space using a point convolution network. As we are going to explain, this design corresponds to setting the number of components $N_{C}$ to 1.

In this basic approach, the same weights describe all the parts of the target shape and struggle to represent local regions with very distinct geometries. Increasing the generator’s complexity, such as the number of convolutions and channels, is not enough to describe the complicated global geometry. The obvious solution is to independently process different regions of the target shape using different weights. At the same time, the intuitive patching approach proposed in [22] leads to the drawbacks explained in Section 2.4.

We introduce an alternative to the patching approach that ensures a smooth transition between local representations and can still run in parallel. We model the target point cloud as a mixture model of shape distributions that we call components. Each component is a Euclidean projection of the target shape that focuses on a different region (Figure 3). The weights of the mixture model indicate which components have the most influence on each point of the output cloud. The component and attention blocks encapsulate the proposed approach.

The component block forms $N_{C}$ projections of $\left\{x_{i}^{2}\right\}$ into the Euclidean space using $N_{C}$ point convolution networks. We refer to the projections as components and denote them by ${\mathrm{Comp}}_{c}\left(\left\{x_{i}\right\}\right),\;c\in \left\{1,2,\ldots ,N_{C}\right\}$. The attention block yields the interpolating weights $g_{i}^{c}$, representing the probability that a point belongs to the component $x_{i}^{c}$. To calculate $g_{i}^{c}$, we proceed as follows (see also Figure 2). We extract local information by taking the features in the component block immediately before the final linear layer. For each component *c*, we obtain an unnormalized score ${\hat{g}}_{i}^{c}$ using a point convolution with a single output channel. The scores ${\hat{g}}_{i}^{c}$ assess how much the component is relevant for each point, corresponding to a self-attention mechanism. We normalize component-wise ${\hat{g}}_{i}^{c}$ using the Gumbel-Softmax [39], which also adds noise to encourage a more uniform weight distribution:$$g_{i}^{c}=\mathrm{Gumbel}\left({\hat{g}}_{i}^{c}\right).$$

The output position of a point with index *i* is as follows:$$x_{i}=\sum_{c=1}^{N_{C}}g_{i}^{c}{\mathrm{Comp}}_{c}\left(x_{i}\right),\;\;\sum_{c=1}^{N_{C}}g_{i}^{c}=1,$$
where $x_{i}$ is a 3D vector denoting the position of one point of the output point cloud. Note that we do not enforce the components to only focus on specific regions. Instead, this occurs naturally as the model learns to distribute the tasks between them from the early stages of training.

### 3.2. Laplacian Filtering

A generator defines a mapping that samples points from a target shape. As we compute the points in parallel, they are independent. A likely consequence is the formation of small gaps on the surface resulting from sampling errors. We design here a conceptually simple and elegant filter that mitigates this effect, filling in possible gaps on the recovered surface. It benefits both training and inference. During training, it improves the quality of the gradients from the reconstruction loss, letting our model focus on the shape and converge more easily. During inference, it provides a quick postprocessing tool for point cloud generators that enhances the visual rendition and the standard quantitative metrics. In line with the overall goal of this paper, our filter is parallelizable.

#### 3.2.1. Spreading the Points along the Surface

We modify the adjacency matrix with exponential weight in Equation (Equation 1) by removing the threshold on the distance of the points. Instead, we consider only each point’s three closest neighbours. More precisely, let *X* be a point cloud and $\delta_{i}$ be the indices of the three points $\left\{x_{j}\in X|j\in \delta_{i}\right\}$ closest to $x_{i}\in Y$. We define the weight matrix *W* as follows:(4)$$W\left(i,j\right)=\left\{\begin{array}{cc} \exp\left(-\frac{\left|\right|x_{i}-x_{j}{\left|\right|}_{2}^{2}}{2\epsilon }\right)\; & \mathrm{if}\;j\in \delta_{i}\\ 0\; & \mathrm{otherwise}. \end{array}\right.$$

Note that this matrix is generally not symmetric but grants a flexible receptive field that grows when the points are more distant.

We use the weight matrix *W* above in Equation (Equation 2) to define the unnormalized Laplacian Δ. The proposed weight matrix provides its action with the following intuitive interpretation. Given a point $x_{i}\in X$, $\Delta (x_{i})$ becomes a weighted mean value change $\left(x_{j}-x_{i}\right)$ from its closest neighbours (see black arrows in Figure 4).

Traditionally, the Laplacian filter (1−λΔ) takes a positive value of λ to displace a point in the direction of its neighbours and smoothen an irregular surface. At the same time, it also facilitates the formation of clusters. We take the opposite approach by selecting λ=−1, therefore pushing each point away from its closest neighbours (see red arrows in Figure 4).

Formally, the proposed filter nudges a point $x_{i}\in X$ to the following: (5)$$\left(1+\Delta \right)x_{i}=x_{i}+\sum_{x_{j}\in X}W\left(i,j\right)\left(x_{i}-x_{j}\right).$$

Assuming a locally flat receptive field, the proposed filter spreads nearby points along the surface. By discouraging the formation of neighbourhoods of high density, our filter helps cover the target surface more homogeneously. Indeed, the filter performs well when varying the number of neighbours considered as long as the receptive field does not grow too much. Figure 5 shows the effect of the filter on a reconstructed sample.

The value of ϵ has a substantial impact on the filter. Instead of assigning a fixed value, which would not be optimal for all training phases, we calculate it dynamically for each point cloud as half the mean distance between closest neighbours: (6)$$\epsilon =\frac{\sum_{i}\min_{j}\left|\right|x_{i}-x_{j}{\left|\right|}_{2}}{2|X|}.$$

Section 3.2.2 proves that $\epsilon =\frac{\min_{j}\left|\right|x_{i}-x_{j}{\left|\right|}_{2}}{2}$ ensures that the displacement is always larger the closer a point is to $x_{i}$. In practice, we use a mean value over the point cloud to circumvent possible instabilities when $\min_{j}\left|\right|x_{i}-x_{j}{\left|\right|}_{2}$ approaches zero.

#### 3.2.2. Mathematical Motivation

We provide mathematical proof here as motivation for this homogenizing effect of the proposed filter. In particular, we prove that, as long as two points are more than 2ϵ distant, the closer they are, the farther our filter pushes them away. We remind the reader that $W:R^{n}\to R_{\geq 0}$ is a radial function centred in $x_{0}\in R^{n}$ if and only if W(x) solely depends on the distance $d=\left|\right|x_{0}-x{\left|\right|}_{2}$. That is, there exists $\hat{W}:R_{\geq 0}\to R_{\geq 0}$ such that $W\left(x\right)=\hat{W}\left(|\right|x_{0}-{x||}_{2})=\hat{W}\left(d\right)$.

**Definition** **1** (Displacement)**.**
*Let W be a radial function centred in $x_{0}$.*

*We define the displacement $\Delta \left(W\right):R^{n}\to R^{n}$ as follows:*

$$\Delta \left(W\left(x\right)\right)=W\left(x\right)\left(x_{0}-x\right).$$



**Definition** **2** (Rescaling radial function)**.**
*We say that a radial function W centred in $x_{0}\in R^{n}$ rescales a set $X\subset R^{n}$ if the norm of the displacement*

$${\left||\Delta \left(W\right)\left(x\right)|\right|}_{2}=\hat{W}\left(|\right|x_{0}-{x||}_{2})||x_{0}-x{\left|\right|}_{2}=\hat{W}\left(d\right)d$$

*is strictly decreasing in d when restricted to the domain $d=\{||x_{0}-x|{|}_{2}|x\in X\}$.*


**Proposition** **1.**
*Let $X\subset R^{n}\setminus \left\{x_{0}\right\}$ be finite and not include $x_{0}\in R^{n}$. The function*

$$W_{X}^{x_{0}}\left(x\right)=\exp\left(\frac{-\left|\right|x_{0}-x{\left|\right|}_{2}}{2\epsilon }\right),$$

*where $\epsilon =\min_{x\in X}\left(\frac{\left|\right|x_{0}-x\left|\right|}{2}\right)$, is a radial function centred in $x_{0}$ that rescales the set X.*


We refer to Appendix B for a proof of the above proposition.

### 3.3. Quantization and Sampling Strategy

We build on the previous sections by devising a quantization strategy that tailors the architecture in Section 3.1. The perks of the resulting VQVAE model grant us the opportunity for further improvement, especially in the context of parallel computing. In addition to the quantization strategy, we introduce a variational autoencoder whose loss and training procedure are specifically designed to sample the discrete codes of the VQVAE model.

The proposed model overcomes the typical limitations of VQVAEs by integrating a VAE to support its generative inference. The inclusion of the VAE model provides a continuous global latent space and a sampling method that enables parallel computation. Therefore, the compound of the two models acts similarly to a singular improved VAE. Quantization replaces a continuous vector with a vector from a codebook, a finite set indexed by a discrete variable. When training the point cloud autoencoder, our model reshapes the codeword *w*, which is the output of the encoder, into chunks and applies quantization at the chunk level (see Figure 6). It then rearranges the quantized chunks into the quantized codeword $\bar{w}$ and relays it to the generator. Similarly, we quantize the chunks generated by the variational autoencoder (the w-autoencoder) during inference and pass them to the generator to produce random point clouds.

As is common in VQVAEs [19], our codebook is a list of learned vectors with the same embedding dimension of the chunks and we use the Euclidean distance as the embedding distance. Our model presents two differences when compared to [19]. The embedding space has a much lower dimension than in standard VQVAE models and can be covered by fewer learned vectors. The second difference is that the discrete variables represent semantic information that is separate and global instead of aggregate and localized. Because of these reasons, we keep a separate codebook for each code, as also proposed in [17] for related reasons. More precisely, let $N_{e}$ be the number of the codes, $N_{b}$ the size of the codebook and $L_{e}$ the embedding dimension. For each code $k\in \{1,2,\ldots ,N_{e}\}$, the corresponding codebook is $\left\{e_{j}^{k}\in R^{L_{b}}|j\in \left\{1,2,\ldots ,N_{b}\right\}\right\}$.

During training, our model tends to rely on a few latent discrete codes, leaving others unused and limiting its expressiveness. To solve the issue, we redistribute the embeddings that the model did not use every few epochs. The redistribution randomly replaces the embedding with another from the same codebook according to the latter usage percentage in the training dataset and adds Gaussian noise. This solution makes the discrete embeddings more homogeneous and improves the reconstruction.

In its original formulation, VQVAE [19] learns an ancestral sampling of the discrete latent space with an autoregressive model. The choice is intuitive, as it leverages the spatial consistency of an image array. There is no such advantage in our case because the quantized chunks yield global information.

We replace the autoregressive model on the discrete space with a variational autoencoder and a second (continuous) $L_{z}$-dimensional latent space governed by a VAMP prior distribution [26] with $N_{u}$ pseudo-inputs (see Figure 7). The proposed second autoencoder, to which we refer as the w-autoencoder, presents significant differences from the standard formulation of a VAE. Naively encoding the discrete distribution does not take advantage of how the pc-autoencoder uses it and can lead to instabilities. Instead, our w-autoencoder implements the following solutions.

We reuse the quantized vectors $e^{k}=e_{l^{k}}^{k}$ as an embedding of the latent variables in the VAE autoencoder so that discrete latent variables with a similar effect during the point cloud generation are also close in the w-autoencoder. In this way, we limit the propagation of the reconstruction error in the w-autoencoder and make the random generation more robust.

The continuous codeword contains more information than its quantized counterpart and may lead to better generation. Because of this, instead of encoding the embedding $e^{k}$, we use the continuous chunk $f^{k}\approx e^{k}$ outputted by the PC encoder. Therefore, the w-autoencoder learns to quantize the input rather than reconstruct it.

We improve the discrete variable prediction by replacing the reconstruction loss of the embeddings with cross-entropy classification loss. To obtain the estimated probability $p_{j}^{k}=P\left(l^{k}=j\right)$, we take the Euclidean distance between the output of the decoder ${\hat{f}}^{\;k}$ and $e_{j}^{k}$. Then, we reshift and softmax all the distances along the possible values *j*, pushing the reconstructed embedding close to the correct one and away from the others. Formally,
(7)$$L_{\mathrm{CE}}=-\sum_{k}\log\left(p_{l^{k}}^{k}\right),\;p_{j}^{k}=\underset{1\leq j\leq N_{b}}{\mathrm{Softmax}}\left(d_{j}^{k}-\min_{1\leq j\leq N_{b}}d_{j}^{k}\right)\;d_{j}^{k}=\left|\right|{\hat{f}}^{\;k}-e_{j}^{k}{\left|\right|}_{2}^{2}.$$

The w-autoencoder architecture independently encodes and decodes the channels corresponding to the chunks into and from a shared latent variable *z*. Avoiding direct connections between the channels helps reduce the required parameters. To encode a codeword, we expand the input embeddings in a larger dimension, using a convolution with a unitary kernel, i.e., a dense layer shared by all embeddings. We concatenate the outputs and infer the hyperparameters of the continuous latent variable *z* with a linear layer. To decode *z*, we give it as input to a dense layer with separate weights for each reconstructed embedding. Note that we can still compute the decoding in parallel.

Figure 8 illustrates the architecture of the w-autoencoder, complementing Figure 2.

### 3.4. Encoder, Training and Final Loss

The PC encoder used in the experiments is a lighter version of DGCNN [9], where we replace the edge convolutions with point convolutions followed by a local max-graph-pooling as in Foldingnet [20], except for the first edge convolution, which we keep. We calculate the nearest neighbours only for the edge convolution and reuse the same neighbours for the max-graph-pooling to reduce the computational burden. The resulting encoder is significantly faster and almost as performant as a full DGCNN. Notice that the encoder does not influence inference time during generation but we choose the lighter version to reduce training time.

It is possible to train the two autoencoders together, provided that you are stopping the gradients of the w-autoencoder from impacting the pc-autoencoder. In our experiments, we have decided to train the w-autoencoder on an already trained pc-autoencoder. Therefore, we present the losses for the two autoencoders, $L_{\mathrm{pc}}$ and $L_{w}$, separately.

The loss for the pc-autoencoder includes a sum of the Chamfer distance $L_{\mathrm{CD}}$ and the Earth mover distance $L_{\mathrm{EMD}}$, both defined in Section 4.1.1, as reconstruction loss. To these, it adds the embedding loss $L_{\mathrm{Emb}}$ corresponding to the mean square distance between the features of the PC encoder $f^{k}$ and the embeddings of the discrete codes $e^{k}$. This loss helps balance the training of the encoder and the generator, and needs a coefficient to allow for better reconstructions. We find it experimentally. In formulae,
(8)$$L_{\mathrm{pc}}=L_{\mathrm{CD}}+L_{\mathrm{EMD}}+c_{\mathrm{Emb}}L_{\mathrm{Emb}}\;c_{\mathrm{Emb}}=2.$$
The loss for the w-autoencoder is a sum of the Kullback–Leibler divergence $L_{\mathrm{KLD}}$ and the cross-entropy loss $L_{\mathrm{CE}}$ defined in Section 3.3. When using a VAMP prior with pseudo-inputs ${\left\{u_{r}\right\}}_{r\in \{1,\ldots ,N_{u}\}}$ and Gaussian posteriors, the KLD loss $L_{\mathrm{KLD}}$ for a latent variable *z* given a codeword *w* has the following form:(9)$$L_{\mathrm{KLD}}\left(w\right)=D_{KL}\left(N\left(z|w\right),p\left(z\right)\right),\;p\left(z\right)=\frac{1}{N_{u}}\sum_{r=1}^{N_{u}}N\left(z|u_{r}\right).$$

We find experimentally a coefficient to balance the two losses. In formulae,
(10)$$L_{w}=L_{\mathrm{CE}}+c_{\mathrm{KLD}}L_{\mathrm{KLD}},\;c_{\mathrm{KLD}}=3.$$

## 4. Experiments

We present several experiments and ablation studies to validate the various contributions of this paper. These include an assessment of our model’s reconstruction ability and quantification of its random generation quality. The dataset employed for these experiments is Shapenet [24], a widely used collection of CAD models of furniture, vehicles, guns and everyday objects. For the reconstruction and random generation experiments, we utilize two different point cloud versions of this dataset from [23,29].

Section 4.1 defines the metrics necessary to compare our works with the related literature quantitatively. Section 4.2 describes the experiments, specifies the hyperparameters, and reports and discusses the results.

### 4.1. Experimental Metrics

A slight deviation in the experimental design leads to incomparable results. This issue is common in the point cloud reconstruction and generation literature, where different versions of the same dataset, incompatible metrics and eventual postprocessing often appear in concurrent work. To prevent confusion, we clarify all the details of the reported experiments and make them fully reproducible.

#### 4.1.1. Reconstruction Losses for Point Clouds

Similarity measures between point clouds are a challenging topic outside the scope of this work. We consider the two most popular reconstruction losses.

The Chamfer distance (CD) for point clouds *X* and *Y* is defined as follows: (11)$$L_{\mathrm{CD}}=\sum_{x\in X}\min_{y\in Y}||x-{y||}_{2}^{2}+\sum_{y\in Y}\min_{x\in X}||x-{y||}_{2}^{2},$$
while the Earth mover distance (EMD) for point clouds *X* and *Y* of the same size is the following:(12)$$L_{\mathrm{EMD}}=\min_{\phi \in \Phi }\sum_{x\in X}||x-{\phi \left(x\right)||}_{2},\;\Phi =\left\{\phi :X\to Y|\phi \left(X\right)=Y\right\}.$$

The EMD is computationally expensive and related literature uses an approximation of it [29]. In this work, we use the same implementation to help compare the results. We use the pykeops library [40] (version 2.1) instead to calculate the Chamfer distance quickly.

#### 4.1.2. Generation Metrics

The experimental settings for assessing the generation quality are the same as in [28]. For both the Chamfer and the Earth mover pseudo-distances, we consider the three following standard metrics.

The generative metrics are the minimum matching distance (MMD), coverage (Cov) and 1-nearest neighbour accuracy (1-NNA), and are defined as follows.

Let *d* be a pseudo-distance between point clouds, $S_{r}$ be a reference set of point clouds and $S_{g}$ be an equally large group of generated point clouds to evaluate. Furthermore, for a point cloud $x\in S=S_{r}\cup S_{g}$, let the closest neighbour in either dataset be $N_{X}={\mathrm{argmin}}_{Y\in S,\;Y\neq X}d\left(X,Y\right)$. The mathematical definitions of the metrics are as follows:$$\mathrm{MMD}\left(S_{r},S_{g}\right)=\frac{1}{|S_{r}|}\sum_{Y\in S_{r}}\min_{X\in S_{g}}d\left(X,Y\right).$$
$$\mathrm{COV}\left(S_{r},S_{g}\right)=\frac{\left|\{{\mathrm{argmin}}_{Y\in S_{r}}d\left(X,Y\right)|X\in S_{g}\}\right|}{|S_{r}|}.$$
$$1\text{-}\mathrm{NNA}\left(S_{r},S_{g}\right)=\frac{|\left\{X\in S_{r}|N_{X}\in S_{g}\right\}\cup \left\{\left(X\in S_{g}|N_{X}\in S_{r}\right)\right\}|}{|S_{r}\left|+\right|S_{g}|}.$$

The minimum matching distance is low when some generated point clouds are realistic and close to the reference group. Coverage is high when the generated point clouds are varied and represent the reference distribution. The 1-nearest neighbour accuracy detects how indistinguishable the two groups are. It is 1 when they are easily distinguishable and 0.5 when they are impossible to separate.

Following the experimental setup in [28], we consider d=CD or d=EMD. $S_{r}$ is a point cloud randomly chosen from the validation dataset. When assessing the baseline, $S_{g}$ is a point cloud randomly extracted from the training dataset. Furthermore, we normalize the generated and extracted samples to fit the unit sphere as in [17], which makes the clouds more entangled and generally lowers the metrics.

#### 4.1.3. Hardware Specifications for Performance Evaluation

We calculate the computation time on a single NVIDIA GeForce RTX 3080 Ti GPU. Even though the entire generative model is parallelizable, our implementation in Pytorch 2.1 [41] calculates the components sequentially. An entirely parallel version of the model runs into time overheads that, with the present library and available hardware, exceed the benefits of parallel computation. Instead, we restrict computation time by limiting the number of components, especially for the generative model. Our code is available in [42].

### 4.2. Experimental Results

This section reports the descriptions and outcomes of two standard experiments commonly used as validation in the related literature. The first experiment tests the reconstruction potential of the proposed generator without any quantization or regularization of the latent space, comparing the effectiveness of the proposed architecture and filter with existing ones for plain point cloud reconstruction. In the second experiment, we assess the quality of the random generation given by the proposed double encoding and its inference time, and compare them with related work. Furthermore, we add one experiment that demonstrates our model’s topological flexibility and qualitative results from conditional generation. A discussion of the experiment ends the section.

For every one of the following experiments, we provide the optimal hyperparameters. The number of hidden features in the architecture, as shown in Figure 2 and Figure 8, are the same in each experiment.

We report the sizes of the hidden dimensions used in the proposed model in Table 1 and note that we select 8 as the temperature parameter for the Gumbel-Softmax.

#### 4.2.1. Reconstruction of Benchmark 3D Models

This experiment consists of autoencoding of the version of the Shapenet dataset [24] that [23] used and it closely follows [23]. In this experiment, we skip the quantization of the codeword and we give the generator the continuous output of the pc-encoder. Each point cloud is centred and has a unit radius. At runtime, the point cloud is randomly sampled without replacement to a size of 2500 points and encoded in a codeword of dimension $L_{w}=1024$. The primary metric is the Chamfer distance of the input point cloud with the reconstructed encoding.

We present a minimal version of our model with only one component (PCGen), a larger one with eight components (PCGen_8_) and one with four (PCGen_4_) as a compromise between the lightness of the former and the performance of the latter, which is the version we select for the generative model in later experiments.

All three versions follow the same training scheme. We select the Chamfer distance as a loss, mini-batches of 16 and the AdamW [43] optimizer with $1.0\times 10^{-6}$ as the coefficient for weight decay. The learning rate starts at $5.0\times 10^{-4}$ and progressively decays following a cosine schedule until it reaches $1.0\times 10^{-6}$ at epoch 250. After that, we train the models for 100 more epochs. We did not adopt augmentation strategies besides resampling.

In Table 2, we report the evaluation of the proposed models together with the results published in [23] of their best-performing models: AtlasNet with point translation (AtlasNetP_10_) and AtlasNet with patch deformation (AtlasNetD_10_), both using 10 patches.

To stress that the gain in performance is mainly due to our decoder, we reimplement the two mentioned versions of AtlasNet and train them using our encoder (defined in Section 3.4) and the same training scheme. We train in the same way the decoders from FoldingNet [20] and TearingNet [21], whose published results are not comparable to ours because of different experimental settings. Following their original article, we build TearingNet [21] on the already trained FoldingNet. Since the memory of our GPU constrains the size of the mini-batches, and all the other training hyperparameters are standard and robust, we only tune the initial learning rate for each model and let it decay as described above. For both versions of AtlasNet, we find that the optimal learning rate is $5.0\times 10^{-5}$, for FoldingNet it is $1.0\times 10^{-5}$ and for TearingNet it is $5.0\times 10^{-6}$.

We include the resulting evaluations in Table 2. The results show that the simplest version of our model improves on previous work on all metrics, offering superior reconstructions at a cheaper computational cost, and that using more components enhances the performance on reconstruction even further. We set the state of the art for a point cloud generator with PCGen_8_. The quality of the reconstructions in Figure 9 confirms the quantitative results.

The success of PCGen is partly due to its improved architecture and partly due to the proposed Laplacian filtering. We support the previous sentence by including two ablation studies.

In the first ablation study, we report the impact of our proposed Laplacian filtering in Table 3. To evaluate it, we train a version of the proposed decoder PCGen with the same training scheme but without the final filtering. During testing, we test it with (PCGen+F) and without (PCGen−F) the filter. We compare them with PCGen, which used the filter during training and evaluation.

The results register a clear improvement over the CD and EMD metrics, and corroborate the benefits of using the filter even during the training phase.

Our filter is ineffective with AtlasNet with point translation, FoldingNet and TearingNet because the deterministic structures already ensure that the points are evenly distributed along the surface. Instead, it works well when applied to AtlasNetD_10_ during evaluation (AtlasNetD10+F in Table 3). Using the proposed filter in the training of AtlasNetD_10_ leads to convergence issues.

The second study (see Table 4) investigates one by one our model architectural novelties by training and evaluating three more standard versions using the same training scheme. These decoders will not use filtering because the ablation focuses on the architecture. The base version (PCGen−FSM) is the closest to AtlasNet with patch deformation with only one patch (AtlasNetD). The support of the distribution of the initial position of the points is a three-dimensional sphere instead of a hypersphere, therefore, a bi-dimensional surface as in AtlasNetD. Furthermore, this version uses concatenation analogously to AtlasNetD, even though PCGen∖FSM maps them in a larger space to better compare with the final version. The two decoders have roughly comparable metrics. The other two versions add the proposed initial sampling (PCGen∖FM) and mixing layer (PCGen−FS), respectively. Both the novelties contribute to the performance, with the mixing layer having the most sizeable effect. We bring additional evidence of the flexibility of our generator with a topologically non-trivial dataset in Section 4.2.2.

#### 4.2.2. Reconstruction of More Complex Topologies

We introduce an experiment that demonstrates that the proposed generator/decoder deals seamlessly with complex topologies. To show that, we rely on visual estimation rather than metrics. We select FoldingNet [20] and TearingNet [21] for baselines with topological limitations.

We create an artificial dataset to prove that the proposed model can handle complex topologies, providing evidence for its overall robustness. The dataset consists of point clouds sampled from flat cylinders with holes that we refer to as pierced coins. The number and position of the holes are random, but they cannot be more than three. The holes are squared with a side length twice the coin’s thickness. To pierce the coin, we consider two squares perpendicularly placed on its sides. We remove the points on the squares and rearrange them on the four sides that connect them. In this way, the pierced coin is a topological surface equivalent to a connected sum of as many tori as the number of holes. We create at runtime 16,384 coins for each epoch and sample 4096 points from each. We train FoldingNet, TearingNet and the proposed PCGen, without filter and with only one component, using the same settings as in the reconstruction experiment. Figure 10 gives a quality assessment of the reconstructions.

#### 4.2.3. Random Generation of Point Clouds

The random generation experiment employs three datasets from the classes “airplane”, “chair” and “car” of the version of Shapenet [24] in [29]. Following previous work [17,28,29,30,44], testing is on the provided validation datasets, which have a different distribution of shapes from the training ones, particularly for the “airplane” class.

First, we compare the proposed VQVAE reconstruction with other generative autoencoders and CanVAE [17] in particular. To replicate their experimental setup, we sample 2048 points with replacement from each point cloud and we normalize them as in the previous experiment. During testing, we de-normalize the reconstructions to their original size and measure the reconstruction error with a different sampling of the input point cloud.

We use the version of our model with four components; we reduce the codeword length $L_{w}$ to 256 and we divide it into $N_{e}=64$ chunks, corresponding to the discrete codes. The codebook of each code has $N_{b}=16$ embeddings of dimension $L_{e}=4$. The training scheme is the following. We opt for the Adam [43] optimizer, batches of 16 and a learning rate of $5.0\times 10^{-4}$ with cosine decay until $5.0\times 10^{-6}$ at epoch 900, with 1000 total epochs.

Table 5 shows that the proposed probabilistic model offers higher-quality reconstruction than previous work (ShapeGF [28], PF [29], DPM [45] and CanVAE [17]). We remark that the discrete latent space of the proposed model is smaller than the one in CanVAE, having only 64 codes against 256 with a codebook size of 16 against 50 per code.

After that, we train the proposed w-autoencoder using the discrete codes of the pc-encoder. We learn $N_{u}=400$ pseudo-inputs and a $L_{z}=16$-dimensional latent space. We use the Adam [43] optimizer, batches of 128 and a learning rate of $1.0\times 10^{-3}$ with cosine decay until $1.0\times 10^{-5}$. The training epochs with learning rate decay are 900. We train all three models for an additional 100 epochs with the lowest learning rate.

The generation metrics defined in Section 4.1.2 fluctuate from run to run, even when using the baseline. Because of that, unlike previous work, which only displays the result of a single run, we compute the metrics of our model for 10 runs and report the mean and the best value for each metric. The metrics indicate in Table 6 that, on average, our model produces generated point clouds close to or better than the reported state of the art. The table reports the results for the models in Table 5, PointGrow [32], SP-GAN [31], PVD [44] and SetVAE [27]. Non-cherry-picked example results from the proposed models (see Appendix C) show that the w-decoder coherently samples hidden components that the generator embodies in sharp shapes. Furthermore, we have superior results compared to SetVAE [27], the only other model with a comparably fast inference [17].

We compare the inference time of the two models in Table 7. While having fewer parameters and considerably fewer MACs, SetVAE has bottlenecks that give the proposed generated model a slight edge in inference speed.

#### 4.2.4. Latent Space Exploration

The VAMP prior distribution stores a list of encodings of representatives of the training data called pseudo-inputs. The model uses these encodings as the Gaussian means of a mixture model. There is the theoretical risk that the model variance is too marginal to affect the output. In this case, our model would not learn a proper interpolation of the dataset distribution and still have good generative metrics. We show some non-cherry-picked samplings that suggest that this type of overfitting does not happen. We fix a pseudo-input and repeatedly use its encoding and its variance to generate new samples. Figure 11 visually demonstrates that its variance is significant enough to allow for variations and that the model has learned an expressive latent local neighbourhood. In our experience, the model exhibits analogous behaviour for all the pseudo-inputs and models.

Our model is the first one, to our knowledge, that can encode a point cloud dataset into 32 global variables and, at the same time, generate sharp images. Having few global variables opens the doors for opportunities in data exploration, disentanglement and anonymization. The visual results appear to confirm the potential of our approach. Figure 12 shows three directions in the latent space with a clear semantic meaning. In our experience and in line with previous work on VAE, the directions have a similar meaning even when starting from a different area of the latent space.

#### 4.2.5. Discussion

Our model achieves state-of-the-art performance according to the standard metrics for generation. Its success is mainly due to its ability to encode complex shapes. The innovations in the architecture and the proposed filtering granted an edge in reconstruction not only versus similar autoencoder architecture but also versus other autoencoder-based generative methods. Figure 11 and Figure 12 illustrate that random and guided generation express the semantics globally without any mismatch between the parts. In our experience, the components blend seamlessly, remaining undetectable. Another important reason for our model’s excellent performance is the expressiveness of the proposed probabilistic approach. Even though it has a few continuous global variables like a regular variational autoencoder, the combination of the VAMP prior with the quantization of the VQVAE effectively combats the typical posterior collapse that affects variational autoencoders. Figure 11 shows that our model offers local semantic changes. The portion of the latent space, which is the same in every row, seems connected to the type of object (a table chair, an office chair or a bench). Instead, the local variance is more connected to the variability within the object type. This observation suggests that the model can interpolate different designations and designs. Figure 12 shows the direct consequence of the previous observation: we can identify directions connected to semantic changes, such as chair thickness, and navigate through the latent space to generate samples with the desired characteristics.

Overall, the results of our experiments proved that the fast decoders we refer to as generators can be backbones for high-quality generative models with potential in time-sensitive applications and rapid prototyping. Although promising, our model suffers from the following limitations. We achieve quick inference thanks to our model’s parallelization of the computation. The raw computation is still very high. This drawback currently limits our model to applications where high-performing hardware is available. A future direction for a lighter version of our model is to sparsify the attention weight and calculate it directly from the quantized encoding. In this way, the components would only process the points they will impact. Another limitation is that the latent space does not have a convex form and interpolating two specific shapes in the latent space may result in a meaningless sampling. This limitation is due to the VAMP prior distribution in the VAE.

## 5. Conclusions

Current generative models for point clouds produce realistic and varied samples. We introduce a model that can carry that out for real-time applications. The proposed generator combines the merits of high-dimensional sampling, a tensor product for code mixing, per-point attention on different components and a filter that spreads the point more homogeneously. Our double encoding allows parallel inference and global latent variables that the current vector-quantized autoencoder lacks. Together, they assemble detailed features in a realistic point cloud. Future work will investigate whether these three separate contributions also successfully apply to different probabilistic models or data.

## Figures and Tables

**Figure 1 sensors-24-01414-f001:**
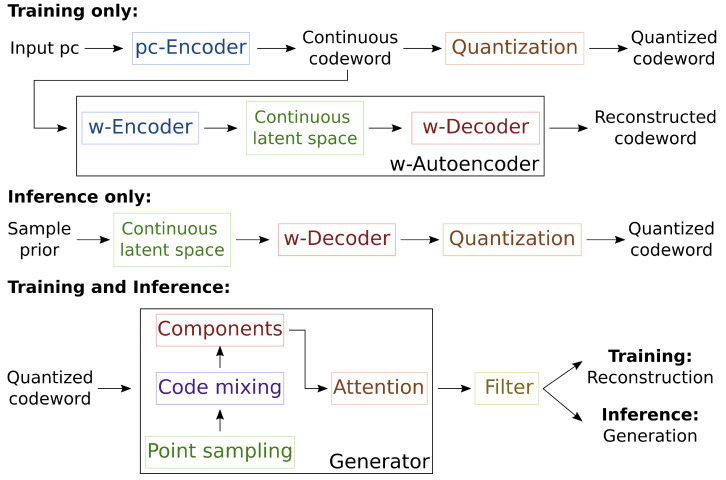
High-level scheme of the model. During training, it encodes the point cloud input in a continuous codeword, which it quantizes. From the quantized codeword, it reconstructs the input with the proposed generator, organized into four blocks (point sampling, code mixing, components and attention) and filter. It also encodes and reconstructs the continuous codeword with the introduced w-autoencoder. During inference, it samples the w-encoder latent space’s prior distribution to generate a quantized codeword and processes it as before to point cloud generation.

**Figure 2 sensors-24-01414-f002:**
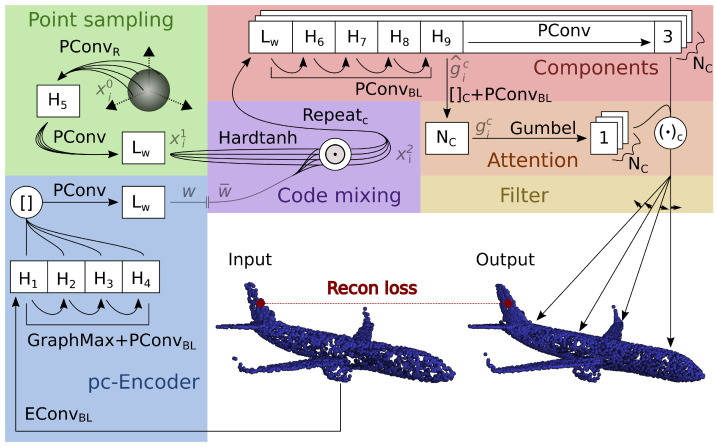
PCGen autoencoding architecture. The pc-encoder compresses the input in a codeword *w* of dimension $L_{w}$. The generator mixes the points from the hypersphere with the quantized codeword $\bar{w}$ and independently processes them into ($N_{C}$) components weighted with self-attention. Finally, the filter rearranges the points more evenly on the target surface. Blocks show the dimensionality of the data at different stages of processing. The hidden dimensions’ sizes $H_{1},H_{2},\ldots ,H_{9}$ are hyperparameters of the model. The suffix after edge and point convolutions (EConv, PConv) specifies when they are followed by batch normalization (B), and by ReLU (R) or LReLU (L) activations. Brackets stand for concatenation, ‖ for quantization, ⊙ for element-wise multiplication and ${\left(\cdot \right)}_{C}$ for dot product along the components. Best viewed in colour.

**Figure 3 sensors-24-01414-f003:**
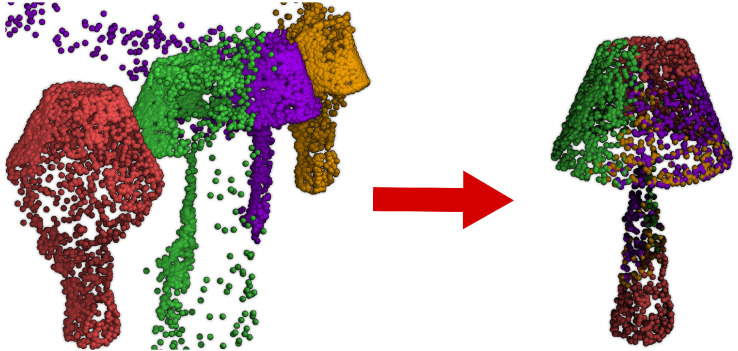
An illustration of the proposed attention mechanism. Four components of a lamp on the left. On the right, the reconstructed lamp. The points’ colour is the same as the component that had the larger impact on them. Each component focuses on a few areas, leaving the others partially or entirely untrained.

**Figure 4 sensors-24-01414-f004:**
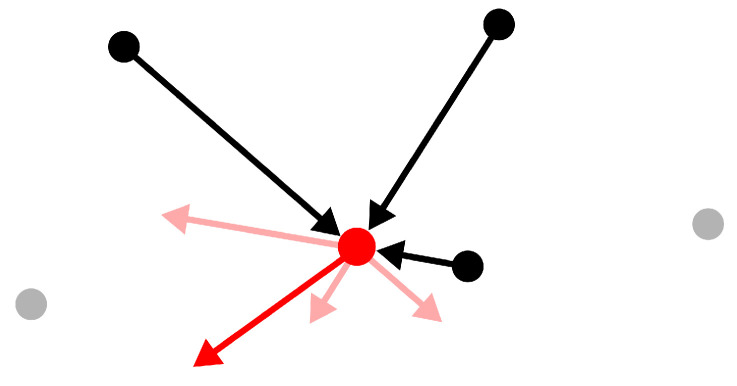
A sketch showing the action of the filter on the target point in the centre (not to scale). The black arrows are the vector distances between the target and its three closest points. Each arrow pushes the target point along its direction; the corresponding displacement (in light red) is inversely proportional to its length. The final displacement of the target point (in bright red) is the vectorial sum of the displacements caused by the neighbouring points.

**Figure 5 sensors-24-01414-f005:**
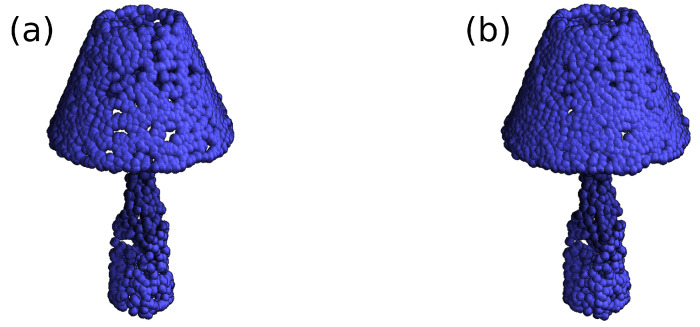
A demonstration of the effect of the filter on the lamp in Figure 3. Subfigure (**a**) shows the lamp’s initial reconstruction before the application of the filter, while subfigure (**b**) displays the final reconstruction after the filter has been applied. The final reconstruction has a smoother appearance with reduced gaps on the surface.

**Figure 6 sensors-24-01414-f006:**
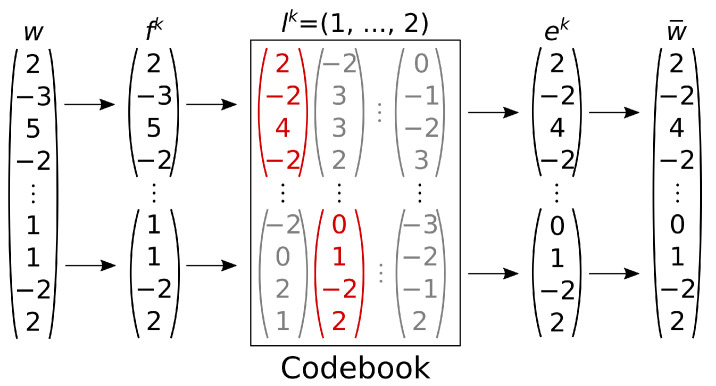
A numerical example of quantization during the forward pass of the pc-autoencoder. In red, the vectors from the codebook closest to the chunks. In gray, the unused vector from the codebook.

**Figure 7 sensors-24-01414-f007:**
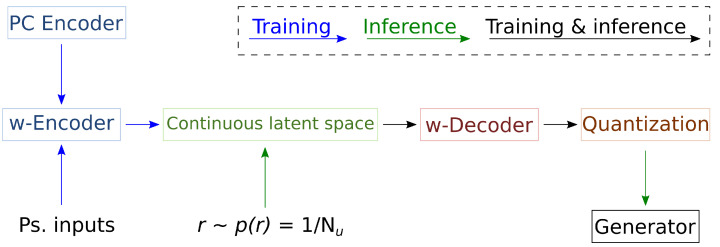
A high-level scheme of training and inference of the proposed w-autoencoder. During training, the autoencoder tries to reconstruct the quantization of the input codeword. It also learns a VAMP prior by using pseudo-inputs. During inference, it randomly selects a pseudo-input and samples and decodes its encoding. The resulting quantized codeword is handed to the generator, which will convert it into a random point cloud.

**Figure 8 sensors-24-01414-f008:**
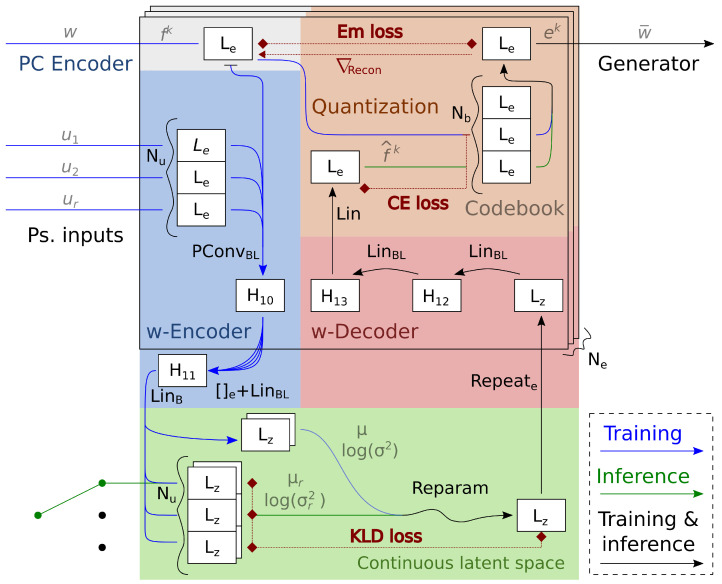
Quantization and w-autoencoder in PCGen. At the top, we reshape the codeword *w* at a chunk level ($L_{w}=N_{e}\times L_{e}$). Quantization replaces chunk $f^{k}$ (top left) from a codeword with an element of the codebook $e^{k}$ (top right). The red dotted arrow indicates the transfer of the gradients from the reconstruction loss. The w-autoencoder compresses the codeword in a continuous latent space. The CE loss encourages the reconstructed chunk ${\hat{f}}^{k}$ to have the same quantization as $f^{k}$ but does not affect the input codeword (see the transversal bar before the w-encoder). “Reparam” is short for the reparameterization trick. Generation starts with uniformly sampling the pseudo-encodings. We reuse the notation of the previous figures and add Lin for the linear layer and the dimensional hyperparameters $H_{10}$, $H_{11}$, $H_{12}$ and $H_{13}$.

**Figure 9 sensors-24-01414-f009:**
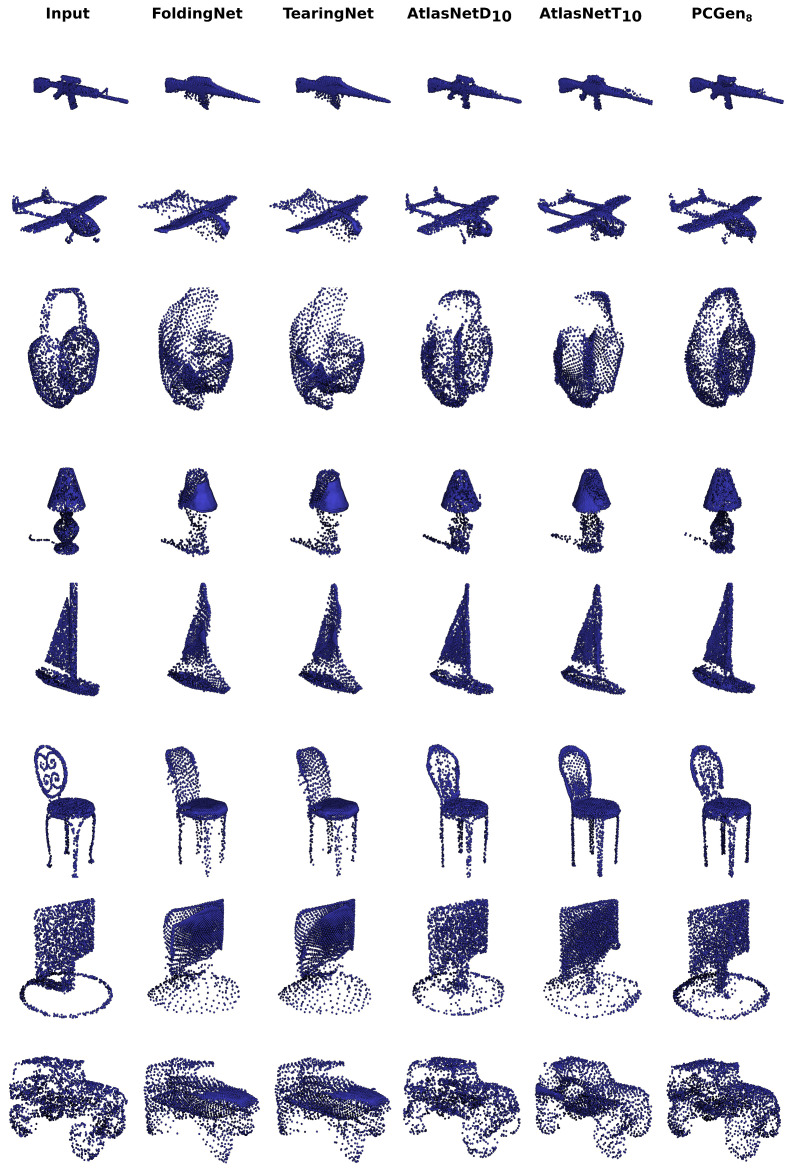
Reconstructions from the first experiment. We show particularly challenging samples to assess the generator’s abilities visually. TearingNet and FoldingNet struggle with high-curvature surfaces (see the corner of the screen). AtlasNet with patch deformation gives reconstructions similar to but less definite than our model (see the lamp’s body). Note also the gaps in the reconstructed car affect their model more than ours, which uses the proposed filter. AtlasNet with point translation has fewer gaps but struggles with complex shapes like the firearm handle or the headphones. Of all the chair reconstructions, ours is the only one that recovers the oval shape of the back.

**Figure 10 sensors-24-01414-f010:**
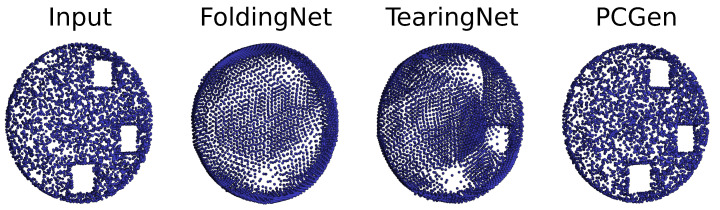
FoldingNet does not reconstruct the hole correctly but leaves a low-density area where the holes should be instead. TearingNet tears the coins where the holes should be but still relies on the folding operation to approximate the sides of the hole. PCGen does not show any artefacts.

**Figure 11 sensors-24-01414-f011:**
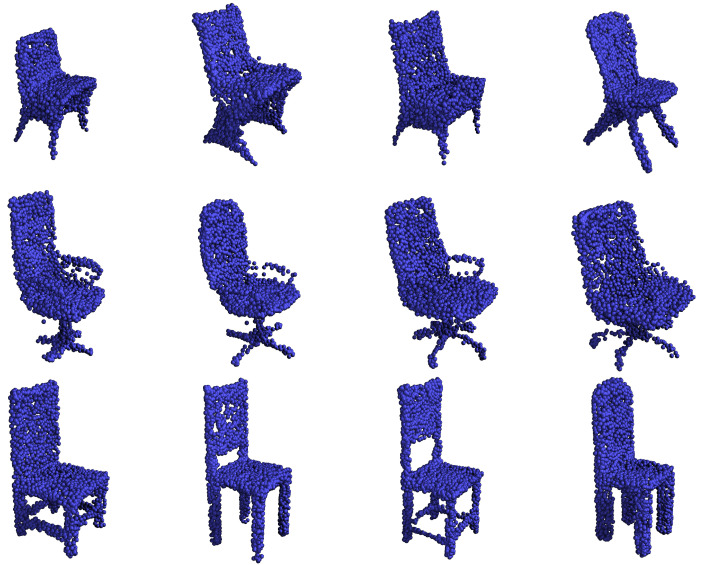
Each row shows randomly generated chairs from the same pseudo-input encoding. Despite some similarities, each pseudo input can result in generated chairs that are very different from each other.

**Figure 12 sensors-24-01414-f012:**
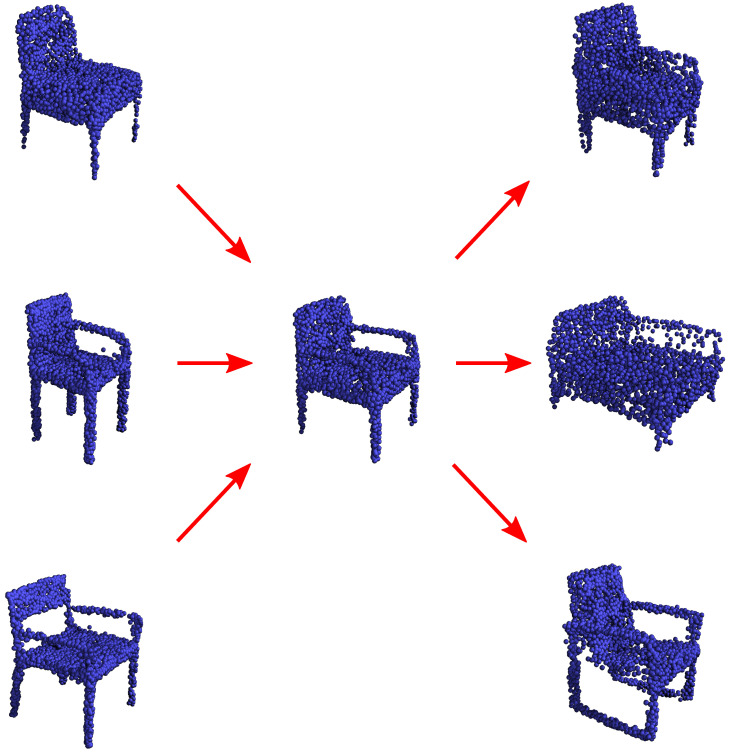
An illustration of semantic directions in the latent space. The generated armchair in the centre is the starting point. One direction, going from the upper left to the lower right in the image, modifies the armchair’s arms and legs. Another direction, going from left to right, relates to the armchair’s width. Finally, the direction from the lower left to the upper right determines how thick the body of the armchair is.

**Table 1 sensors-24-01414-t001:** Hidden dimensions used in all the experiments (see Figure 2 and Figure 8).

$H_{1}$	$H_{2}$	$H_{3}$	$H_{4}$	$H_{5}$	$H_{6}$	$H_{7}$	$H_{8}$	$H_{9}$	$H_{10}$	$H_{11}$	$H_{12}$	$H_{13}$
64	64	128	256	256	512	512	512	64	32	2048	128	64

**Table 2 sensors-24-01414-t002:** Test results for the autoencoder. The number of parameters (Params) and multiply–accumulate operations (MACs), and forward latency (Latency) only apply to the decoder part of the model and are computed on a batch of 16. The Oracle gives a different sampling (with replacement) of the input point cloud. Best results in bold.

Model	Params (M)	MACs (G)	Latency (ms)	CD ($\times 10^{3}$)	EMD ($\times 10^{2}$)
AtlasNetP_10_ [23]	-	-	-	1.21	-
AtlasNetD_10_ [23]	-	-	-	1.39	-
FoldingNet	1.59	63.14	10.63	1.685	11.6
TearingNet	3.23	192.06	24.0	1.569	10.7
AtlasNetD_10_	17.6	70.23	17.1	1.286	8.06
AtlasNetP_10_	17.4	69.52	13.25	1.135	10.61
PCGen (ours)	1.35	53.91	8.20	1.124	6.94
PCGen_4_ (ours)	4.61	183.74	22.00	1.090	6.64
PCGen_8_ (ours)	8.95	356.9	43.23	1.0664	6.44
Oracle	-	-	-	1.037	5.36

**Table 3 sensors-24-01414-t003:** Performance with and without (−F ) the final filtering. We use the notation +F when the filter is not applied during training but only when evaluating. Metrics as in Table 2. Best results in bold.

Model	Params (M)	MACs (G)	Latency (ms)	CD ($\times 10^{3}$)	EMD ($\times 10^{2}$)
AtlasNetD_10_	17.6	70.23	17.1	1.286	8.06
AtlasNetD10+F	17.6	70.23	17.60	1.204	8.11
PCGen−F	1.35	53.91	7.20	1.204	7.09
PCGen+F	1.35	53.91	8.20	1.141	6.961
PCGen	1.35	53.91	8.20	1.124	6.94

**Table 4 sensors-24-01414-t004:** Architecture ablation results and comparison with AtlasNetD. The letters after − denote which of our contributions we removed: F for the filter, M for the mixing layer and S for the high-dimensional sampling. Metrics as in Table 2. Best results in bold.

Model	Params (M)	MACs (G)	Latency (ms)	CD ($\times 10^{3}$)	EMD ($\times 10^{2}$)
AtlasNetD	1.76	35.12	5.38	1.397	7.94
PCGen−FSM	1.87	74.75	8.71	1.397	7.99
PCGen−FM	1.89	74.88	8.72	1.335	7.35
PCGen−FS	1.35	53.78	7.18	1.255	7.42
PCGen−F	1.35	53.91	7.20	1.204	7.09

**Table 5 sensors-24-01414-t005:** Reconstruction metrics with experimental settings as in [28]. Previous work as reported in [17]. Best results in bold.

Dataset	Metric	PF	ShapeGF	DPM	CanVAE	PCGen	Oracle
Airplane	CD	1.208	0.966	0.997	0.889	0.8716	0.837
	EMD	2.757	2.562	2.227	2.122	2.202	2.062
Chair	CD	10.120	5.599	7.305	6.177	5.507	3.201
	EMD	6.434	4.917	4.509	4.218	4.006	3.297
Car	CD	6.531	5.328	5.749	5.050	4.836	3.904
	EMD	5.138	4.409	4.141	3.614	3.614	3.251

**Table 6 sensors-24-01414-t006:** The values for the other models are as reported in [17] and refer to single runs. The arrows ↓↑ are reminders of when lower or higher is better. The baseline train samples point clouds from the training dataset. Best two results in bold (our best run is excluded for fairness).

		MMD ↓		COV(%) ↑		1-NNA (%) ↓	
Category	Model	CD $\times \;10^{3}$	EMD $\times \;10^{2}$	CD	EMD	CD	EMD
Airplane	**PointGrow**	3.07	11.64	10.62	10.62	99.38	99.38
	**ShapeGF**	1.02	6.53	41.48	32.84	80.62	88.02
	**SP-GAN**	1.49	8.03	30.12	23.21	96.79	98.40
	**PF**	1.15	6.31	36.30	38.02	85.80	83.09
	**SetVAE**	1.04	6.16	39.51	38.77	89.51	87.65
	**DPM**	1.10	7.11	36.79	25.19	86.67	90.49
	**PVD**	1.12	6.17	40.49	45.68	80.25	77.65
	**CanVAE**	0.83	5.50	45.67	44.19	63.45	71.60
	**PCGen_mean_**	0.90	5.67	44.32	43.83	70.90	71.67
	**PCGen_best_**	0.88	5.59	46.67	45.93	69.01	68.64
	Train	0.97	5.80	45.68	46.67	71.36	69.63
Chair	**PointGrow**	16.23	18.83	12.08	13.75	98.05	99.10
	**ShapeGF**	7.17	11.85	45.62	44.71	61.78	64.27
	**SP-GAN**	8.51	13.09	34.74	26.28	77.87	84.29
	**PF**	7.26	12.12	42.60	45.47	65.56	65.79
	**SetVAE**	7.60	12.10	42.75	40.48	65.79	70.39
	**DPM**	6.81	11.91	43.35	42.75	64.65	69.26
	**PVD**	7.65	11.87	45.77	45.02	60.05	59.52
	**CanVAE**	7.37	11.75	45.77	46.07	60.12	61.93
	**PCGen_mean_**	6.92	11.34	45.67	48.97	60.34	60.48
	**PCGen_best_**	6.79	11.28	47.43	50.75	58.45	58.91
	Train	7.636	11.88	48.34	49.09	55.89	58.84
Car	**PointGrow**	14.12	18.33	6.82	11.65	99.86	98.01
	**ShapeGF**	3.63	9.11	48.30	44.03	60.09	61.36
	**PF**	3.69	9.03	44.32	45.17	63.78	57.67
	**SetVAE**	3.63	9.05	39.77	37.22	65.91	67.61
	**DPM**	3.70	9.39	38.07	30.40	74.01	73.15
	**PVD**	3.74	9.31	43.47	39.49	65.62	63.35
	**CanVAE**	3.31	8.89	41.76	47.72	55.68	57.81
	**PCGen_mean_**	3.34	8.87	45.25	47.75	59.40	58.25
	**PCGen_best_**	3.31	8.83	48.01	50.85	57.81	56.67
	Train	3.74	9.38	53.12	47.16	52.70	58.52

**Table 7 sensors-24-01414-t007:** Inference time and MACs to generate a batch of 32 point clouds. The generator takes in a smaller codeword than the previous examples and is, therefore, smaller. Note that for the total inference time we have to add the time for sampling from the VAMP prior to the duration of the two decoders. Best results in bold.

Model	Part	Params (M)	MACs (G)	Latency (ms)
**SetVAE**	Inference	0.39	13.63	31.27
**PCGen**	Generator	2.84	185.07	24.25
	w-Decoder	0.43	0.01	0.89
	Total Inference	3.27	185.08	27.17

## Data Availability

Data are contained within the article.

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
