# Peer review of "PCGen: A Fully Parallelizable Point Cloud Generative Model"

_sensors, 2024, doi:10.3390/s24051414_

Round 1

Reviewer 1 Report

Comments and Suggestions for Authors

In this work, the authors presented a fully parallelizable vector-quantized variational autoencoder model (VQVAE) that generates high-quality 3D point clouds in milliseconds. The authors should address the following points :

- In the abstract the authors state that "Real-time computing is essential to virtual and augmented reality". It would be better to add more detail, in the introduction part, as to the usefulness of the proposed approach in areas such as augmented reality or other.

- In figure 5, it is better to add the lamp figure before and after applying the filter.

- In lines 349 and 354, the authors cited figure 3.2, which was not found.

- Figures 4 and 7 have not been cited in the text.

- Try to improve the organization of the experimentation part.

Reviewer 2 Report

Comments and Suggestions for Authors

This study proposed a fully parallelizable vector-quantized variational autoencoder model for generating point cloud. By comparing this model with established and concurrent work, the results showed more efficient and higher accuracy. It is interesting and helpful for the point cloud generative model research.

However, there are some issues or questions could be addressed:

1) In the abstract, the important meaning and crucial novel contributions of this study should be explained clearly.

2) In the Section 1 of Introduction, the important meaning and existed problems in previous studies, such as synthesizing 3D objects, surface representation and point cloud encoding efficient, should be indicated in detail.

3) In the Section 2 of Related Work, the authors should organize the subsection and content logically. The reconstruction losses for point clouds may not be a review of the related work.

4) In the Section 3 of Methods, the method architecture or flowchart of Figure 1 is not easily understood, which should include a complete flow of data and algorithms. 

5) In the Section 3 of Methods, are all the used components proposed by the authors? If not, it had better described the more important or novel modules in detail and other commonly used in short.

6) In the Section 4 of Experimental results, the data sources and experiment areas should be listed and explained. What is the meaning of "Hidden dimensions" in Table1? Is it data source?

7) In the Sectin 4 of Experimental results, the authors should analyze the advantages and disadvantages of this study deeply.  It should explain the subfigures in Figure 11.

8) Some spelling or syntax mistakes should be carefully checked, such as the number of titles and their case sensitivity, etc.

9) Some related studies should be included and cited in this study, such as: Lin, et al, 2022, https://doi.org/10.1109/ACCESS.2022.3196388; etc.

Comments on the Quality of English Language

Minor editing of English language is needed.

Reviewer 3 Report

Comments and Suggestions for Authors

This paper introduces a method for point cloud generation based on Vector Quantised Variational Autoencoder (VQVAE). The method employs a flexible sampling strategy, enabling parallel generation processes.

The experimental section clearly demonstrates the effectiveness of this method, showing superior performance compared to other approaches.

Author Response

We thank the Reviewer for thoroughly reviewing and correctly summarizing our paper.

Round 2

Reviewer 2 Report

Comments and Suggestions for Authors

The author has addressed the issues and suggestions related to my last review. I hava no other questions.

Comments on the Quality of English Language

Minor editing of English language is needed.